# Boosting Semi-Supervised Scene Text Recognition via Viewing and Summarizing

**Yadong Qu, Yuxin Wang,**[*] **Bangbang Zhou, Zixiao Wang, Hongtao Xie, Yongdong Zhang**
University of Science and Technology of China, Hefei, China
{qqqyd, bangzhou01, wzx99}@mail.ustc.edu.cn
{wangyx58, htxie, zhyd73}@ustc.edu.cn

## Abstract

Existing scene text recognition (STR) methods struggle to recognize challenging texts, especially for artistic and severely distorted characters. The limitation lies in the insufficient exploration of character morphologies, including the monotonousness of widely used synthetic training data and the sensitivity of the model to character morphologies. To address these issues, inspired by the human learning process of viewing and summarizing, we facilitate the contrastive learning-based STR framework in a self-motivated manner by leveraging synthetic and real unlabeled data without any human cost. In the viewing process, to compensate for the simplicity of synthetic data and enrich character morphology diversity, we propose an Online Generation Strategy to generate background-free samples with diverse character styles. By excluding background noise distractions, the model is encouraged to focus on character morphology and generalize the ability to recognize complex samples when trained with only simple synthetic data. To boost the summarizing process, we theoretically demonstrate the derivation error in the previous character contrastive loss, which mistakenly causes the sparsity in the intra-class distribution and exacerbates ambiguity on challenging samples. Therefore, a new Character Unidirectional Alignment Loss is proposed to correct this error and unify the representation of the same characters in all samples by aligning the character features in the student model with the reference features in the teacher model. Extensive experiment results show that our method achieves SOTA performance (94.7% and 70.9% average accuracy on common benchmarks and Union14M-Benchmark). Code will be available at https://github.com/qqqyd/ViSu.

## 1 Introduction

Scene text recognition (STR) aims to recognize text in cropped text images. As a fundamental task, STR can provide auxiliary information for understanding natural scenes and has wide applications in financial systems, virtual reality, and autonomous driving. Because of the expensive and time-consuming annotation process, most STR methods turn to two commonly used synthetic datasets, MJSynth [14] and SynthText [12]. Despite achieving satisfactory recognition results, they still struggle to perform well in challenging real-world scenarios. We attribute the limitations to insufficient exploration of character morphologies. First, as shown in Fig. 1(a, b, c), synthetic data are almost simple samples, whose character morphologies are significantly different from the artistic and distorted real-world characters. The limitations exhibited by training models using only synthetic data are further amplified on the challenging benchmarks WordArt [45] and Union14M-Benchmark (Union-B) [15]. For example, equipped with a language model, ABINet [10] only reaches 67.4% and 46.0% accuracy on WordArt and Union-B. Second, the model is not robust to the morphological

---

[*]Corresponding author

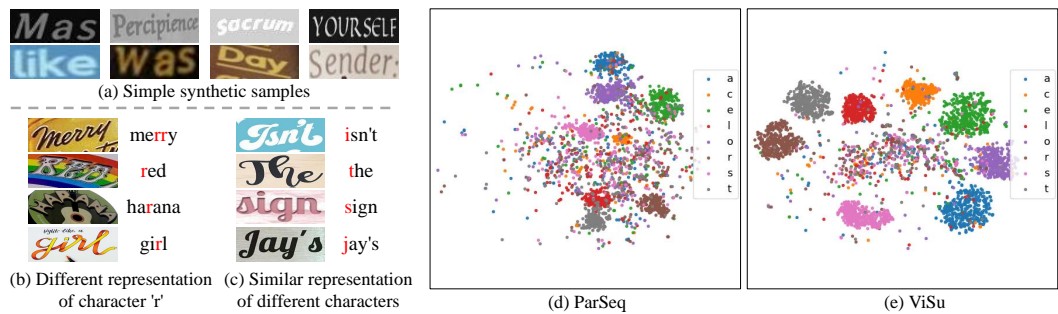

(a) Simple synthetic samples

(b) Different representation of character 'r'

(c) Similar representation of different characters

(d) ParSeq

(e) ViSu

Figure 1: (a) shows some images from synthetic datasets MJSynth and SynthText. (b) and (c) show several challenging test images. (d) and (e) display the visualization of character feature distribution.

changes in characters. As shown in Fig. 1(b, c), due to severe distortion and artistic style, a character can visually differ from others of its category, and on the contrary, characters belonging to different categories can have similar appearances. Fig. 1(d) further shows the distribution of challenging character features in ParSeq [2] from t-SNE [38], where the characters cannot be clearly distinguished.

To compensate for the diversity of character morphologies, some previous methods [1, 10] introduce real unlabeled data by first training a model in the supervised way and generating pseudo-labels. However, it relies heavily on the quality of the pretrained model. The wrong pseudo-label will accumulate the errors and harm the training process. TRBA-cr [54] and Yang *et al.* [47] employ character-level consistency regularization to train the model in a semi-supervised manner and align the sequence output from two views of unlabeled data. Nevertheless, simply assimilating more real unlabeled data and ignoring the diverse character morphologies can lead to improvement on common benchmarks but little superiority on the challenging Union-B. To enhance the robustness to character styles, some methods [44, 20] synthesis the texts with unified styles and map the texts with different writing styles to the invariant features. However, they focus on handwritten mathematical expressions without the diverse orientations and affine transformations common in complicated scene text images.

In this paper, we attribute the bottleneck of the STR model to the **monotonousness of training data** and **sensitivity of the model to character morphologies**. Inspired by the human learning process, we divide the scene text recognition into two steps: viewing and summarizing. Viewing means that when encountering a new language, the first thing is to collect and see as many characters with different morphologies as possible. This also conforms to the principle that deep learning models are data-driven. After seeing enough samples, we can summarize their visual feature commonalities and classify them into different characters. With a small amount of label guidance, we can continuously optimize this cognitive process and distinguish some challenging characters, which is referred to as summarizing. Following this paradigm, we propose our semi-supervised scene text recognition framework by Viewing and Summarizing (ViSu).

To enrich the diversity of character glyphs in the viewing process, firstly, real unlabeled data is similar to practical application scenarios and has a variety of character morphologies. Based on contrastive learning, we adopt the Mean Teacher framework to jointly optimize the model using real unlabeled data and simple synthetic data. Secondly, to further enrich the training data and compensate for the simplicity of synthetic data, we propose an Online Generation Strategy (OGS) to generate background-free samples with challenging styles based on the annotations of the corresponding synthetic data. Compared with the vanilla Mean Teacher framework that simply uses labeled data to train the student model, our method equipped with OGS allows the teacher model to generate guidance of various character styles without the interference of background noise. Thereby, the student model can learn practical features robust to character morphological diversity. Without any human annotation, the model is able to generalize the ability to recognize challenging samples while only training with simple synthetic data, which substantially raises the performance upper bound of the semi-supervised learning framework.

After viewing many characters with various shapes, in order to imitate the summarizing process, the model needs to identify the commonalities of each character and cluster the same characters. Firstly, as shown in Fig. 2, the eight inconsistent representation forms of scene text images caused by reading order and character orientation undoubtedly aggravate the convergence burden. We unify

them into two primary forms based on the aspect ratio to enable the model to extract applicable and discriminative character-level features. Secondly, we theoretically prove that the previous character contrastive loss [45] confuses some character classes and incorrectly encourages the sparsity of the intra-class distribution, thus hindering the optimization of the model. Therefore, a novel Character Unidirectional Alignment (CUA) Loss is proposed to eliminate this mistake and focus on character morphological feature alignment to achieve character clustering within the same class. The images from OGS and weak augmented images, referred to as base images, make it easier to obtain noise-free and glyph-diversified character features that help recognize complex samples. CUA Loss forces the character features under strong data augmentation to be aligned with the base image features from the teacher model, allowing the model to obtain unified features for each category of characters.

The main contributions are as follows: 1) We propose the viewing and summarizing paradigm to conduct the training of the semi-supervised model. Our model can adapt to complex scenarios such as multi-orientation and multi-morphic characters without any human annotation. 2) An Online Generation Strategy is proposed to facilitate model generalization from simple synthetic data to recognize challenging samples, which in turn improves the performance upper limit of the semi-supervised learning framework. 3) A novel Character Unidirectional Alignment Loss is proposed to theoretically correct the formula error of regarding some positive samples as negative samples and enhances the compactness of the intra-class distribution. 4) Our method achieves state-of-the-art performance with an average accuracy of 94.7% on common benchmarks and 70.9% on the challenging Union14M-Benchmark.

## 2    Related Work

### 2.1    Scene Text Recognition

STR methods can be roughly divided into language-free and language-based methods, most of which are trained in a fully supervised manner on synthetic datasets. Due to the simplicity of synthetic data and the diversity of text styles in real scenes, existing methods still have difficulty recognizing complex samples. Some language-free methods [34, 22, 23, 46, 51, 55] design different rectification modules to convert distorted irregular text into regular text to improve recognition accuracy. CornerTransformer [45] proposes to utilize the corner points of artistic text to guide the encoder in extracting useful features from characters. The language-based methods [42, 2, 10, 41, 52, 49] do not rely solely on the visual features of text images. They infer the contextual information of complex characters by introducing additional linguistic knowledge, enabling the model to correct the visual recognition results. MGP-STR [41] proposes a multi-granularity (character, subword and word) prediction strategy to implicitly integrate linguistic knowledge with the model.

### 2.2    Semi-Supervised Learning

As widely acknowledged, deep learning methods require substantial labeled data for effective model fitting. However, acquiring a large amount of high-quality labeled data is typically challenging. Therefore, semi-supervised learning aims to improve model performance by leveraging large amounts of unlabeled data and limited labeled data. Yang *et al.* [48] comprehensively analyses semi-supervised learning methods. They can mainly be categorized into deep generative methods [36, 7, 26], consistency regularization methods [29, 31, 19, 37], and pseudo-labeling methods [3, 8, 21, 4]. CatGAN [36] considers the mutual information between observed samples and their class distribution by modifying the original GAN loss. Temporal Ensembling [19] requires consistent predictive outputs of data under various regularization and data augmentation conditions. Mean Teacher [37] utilizes the exponential moving average (EMA) to update the parameters of the teacher model during training, which tends to generate a more accurate model. Co-training [3] models posit that each sample in the dataset has two distinct and complementary views, and either view is sufficient to train a good classifier.

### 2.3    Semi-Supervised Text Recognition

Some researchers propose integrating semi-supervised learning with text recognition to enhance model recognition performance. Baek *et al.* [1] chooses to pre-train a model only using a small amount of real labeled data, and then further enhances the model performance using Pseudo Label or Mean Teacher with unlabeled data. ABINet [10] proposes an ensemble self-training strategy to

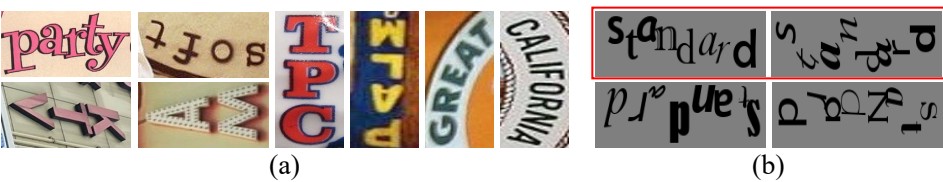

(a)                          (b)

Figure 2: (a) All possible representations of English text images according to character orientation and reading order. (b) The unified representation forms of the word "standard" obtained through Online Generation Strategy. The first row with a red border shows two primary forms, and the second row can be obtained by rotating them 180 degrees.

enhance model recognition performance. However, these two methods require two-stage training, which greatly reduces the training efficiency of the model. Zheng *et al.* [54] is the first to apply a semi-supervised framework based on consistency regularization to the STR task. It designs a character-level consistency regularization to align the characters of the recognition sequence. Gao *et al.* [11] employs well-designed edit reward and embedding reward to assess the quality of generated sequence and optimizes the network using reinforcement learning. Seq-UPS [27], considering that generated pseudo labels may not always be correct, proposes a pseudo label generation and an uncertainty-based data selection framework.

## 3  Methodology

As illustrated in Fig. 3, our framework adopts the Mean-Teacher architecture. The student model and teacher model use the transformer-based encoder and decoder. The parameters of the student model $\Theta_s$ are updated through the backpropagation algorithm, while the parameters of the teacher model $\Theta_t$ are an exponential moving average (EMA) of $\Theta_s$. It can be formulated as $\Theta_t \leftarrow \alpha\Theta_t + (1-\alpha)\Theta_s$, where $\alpha \in [0,1]$ is the smoothing factor. For the viewing process, we introduce not only augmented real unlabeled data but also the Online Generation Strategy (OGS) to obtain samples with challenging glyphs to eliminate the model's sensitivity to character morphology. For the summarizing process, we theoretically analyze the flaw of the existing character contrastive loss and further propose Character Unidirectional Alignment (CUA) Loss to align character features from different views, thus achieving a unified feature representation of multiple-morphic characters. Details will be introduced in the following sections.

### 3.1  Online Generation Strategy

Adopting the semi-supervised framework is to leverage extra real unlabeled data to enrich the character morphologies during the viewing process, but no modification has been made to the existing supervised branch. To address the problem that the labeled synthetic training data is still simple and monotonous, we propose OGS to unify the training process of labeled and unlabeled data and raise the performance upper bound of the semi-supervised framework.

Firstly, as shown in Fig. 2(a), eight inconsistent representations brought about by the diverse character orientations and reading orders undoubtedly increase the difficulty of network convergence. Besides, scene text images with vertical reading orders often have small aspect ratios. When they are resized for input into the STR model, aspect ratios change significantly, resulting in them containing little practical information. Therefore, an image with $height/width > r$ will be rotated, otherwise it remains unchanged, which is referred to as Unified Representation Forms (URF). As shown in Fig. 2(b), unified four representations can be further simplified into two primary forms with left-right reading order. Through URF, we drastically reduce eight representation forms to two primary ones to lessen the learning difficulty and lay the foundation for the following generation process.

Secondly, to enrich the character morphology in the supervised branch, according to the labels of simple synthetic training data, we propose to generate text samples in two primary forms in Fig. 2(b) with the red border. The generated samples are background-free but with random character styles to encourage the model to concentrate on character morphology. Following the characteristics of real samples, all character orientations in one generated sample are the same. OGS samples serve as base images to guide the alignment of the character features in the student model. Unlike previous

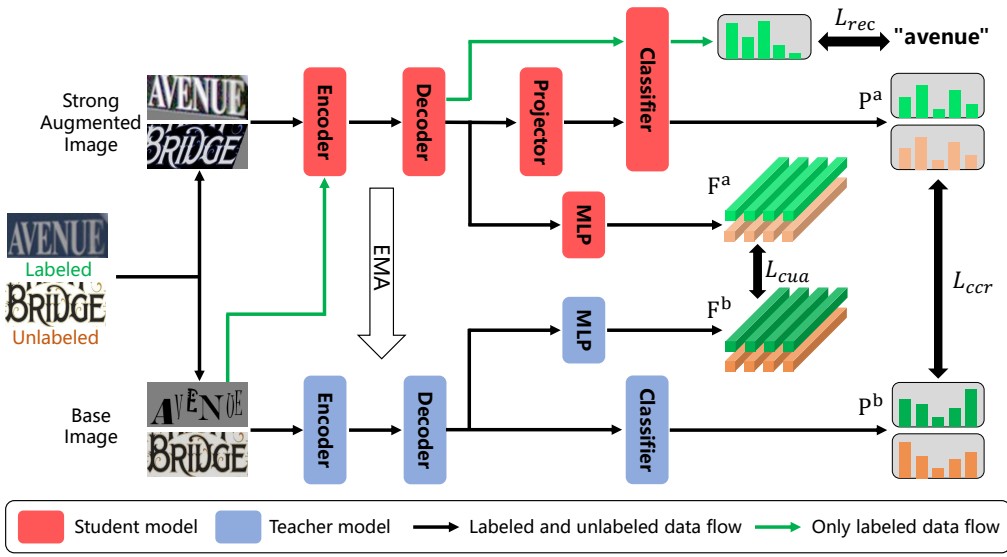

Figure 3: Our framework consists of the student and teacher model. $\mathcal{L}_{rec}, \mathcal{L}_{ccr}, \mathcal{L}_{cua}$ mean recognition loss, character consistency regularization loss and character unidirectional alignment loss. Green and orange stand for labeled and unlabeled data, respectively.

semi-supervised methods [54, 11, 47], our design is based on the following reasons. 1) The goal of a STR model is to obtain character visual features that are not affected by background noise. Because the teacher model is gradient-free and needs to instruct the training of the student model, OGS is expected to generate samples that are noise-free and adaptable to various character morphologies. 2) Due to the existence of word-level annotations and the fact that OGS samples only contain various character features, feeding them to the teacher model is more helpful for the model to learn valuable features for recognizing characters in artistic style or severe distortion. In summary, by exploring the introduction of real unlabeled data and the extension of simple synthetic data during the viewing process, the semi-supervised model can significantly improve the recognition of challenging texts without introducing any human annotation, and break through the performance upper limit.

## 3.2 Character Unidirectional Alignment Loss

After viewing a large number of various characters, the model is supposed to summarize the commonality of each character. As mentioned above, the base images input to the teacher model are more likely to have practical and noise-free character features with diverse glyphs. Because the gradient of the teacher model is stopped, the character features of strong augmented images can be effectively aligned with base images to enhance the robustness of the model to character morphologies and cluster the same characters. Therefore, the CUA Loss is proposed to achieve this goal. A character will calculate the similarity with all the characters in a minibatch. Characters within the same class are supposed to have similar features, while the others should exhibit distinct features. Specifically, the model extracts a sequence of features $\mathbf{F}^a = \{f_1^a, f_2^a, \cdots, f_T^a\}, \mathbf{F}^b = \{f_1^b, f_2^b, \cdots, f_T^b\}$ for augmented and base images, respectively, where $T$ is the maximum recognition length in the decoder. At position $i$, $f_i^a$ and $f_i^b$ are the character features for augmented and base images. $y_i^b$ is the ground truth for labeled data and the prediction from the teacher model for unlabeled data. We define the set for all position $I \equiv \{1, 2, \cdots, B \times T\}$, where $B$ is the batch size. The positive set for character at position $i$ is $P(i) \equiv \{p \in I : y_p^b = y_i^b, S_i^b > \eta_{cua}\}$, where $\eta_{cua}$ is the threshold for confidence score $S_i^b$. For labeled data, $S_i^b$ is 1. For unlabeled data, $S_i^b$ is the confidence score for the whole predicted word $S^b$. The positive set excluding character at position $i$ is $P'(i) \equiv P(i) - \{p \in I : p = i\}$, and the negative set can be represented as $N(i) \equiv I - P(i)$. Our CUA Loss can be formulated as

$$\mathcal{L}_{cua} = -\sum_{i \in I} \frac{1}{|P(i)|} \sum_{p \in P(i)} log \frac{exp(f_i^a \cdot f_p^b / \tau)}{A(i,p)}, \tag{1}$$

$$A(i,p) = exp(f_i^a \cdot f_p^b/\tau) + \sum_{p' \in P'(p)} exp(-f_i^a \cdot f_{p'}^b/\tau) + \sum_{n' \in N(i)} exp(f_i^a \cdot f_{n'}^b/\tau), \quad (2)$$

where $|P(i)|$ means the number of positive characters with $f_i^a$ and $\tau$ is the temperature factor.

It is worth noting that our CUA Loss differs from Character Contrastive (CC) Loss [45] in both motivation and operation. The formulation of CC Loss can be expressed as

$$\mathcal{L}_{cc} = -\sum_{i \in I} \frac{1}{|P'(i)|} \sum_{p \in P'(i)} log \frac{exp(f_i \cdot f_p/\tau)}{B(i)}, \quad (3)$$

$$B(i) = \sum_{p' \in P'(i)} exp(f_i \cdot f_{p'}/\tau) + \sum_{n' \in N(i)} exp(f_i \cdot f_{n'}/\tau), \quad (4)$$

where all the symbols have the same meaning as Eq. 1 and Eq. 2. First, the motivation of CC Loss is to simply cluster the same character in a minibatch, and all images are treated equally. While our CUA Loss works on two types of images: strong augmented images and base images. As stated before, due to the different roles played by the two types of images in our framework and the gradient-free characteristic of the teacher model, all positive characters in base images are targeted for alignment by strong augmented images. Second, CC Loss suffers from a theoretical formula error and incorrectly sparses the distribution of characters belonging to the same category. As shown in Eq. 3, when $f_i$ calculates the similarity with $f_p$, other characters belonging to the same class appearing in the denominator are considered as negative samples by mistake. To be specific, the gradient of the two loss functions can be expressed as

$$\frac{\partial \mathcal{L}_{cua}}{\partial f_i^a} = -\frac{1}{\tau |P(i)|} \sum_{p \in P(i)} \frac{M_1 \cdot f_p^b + \sum_{p' \in P'(p)} M_2 \cdot f_{p'}^b + \sum_{n' \in N(i)} M_3 \cdot f_{n'}^b}{A(i,p)}, \quad (5)$$

$$\frac{\partial \mathcal{L}_{cc}}{\partial f_i} = -\frac{1}{\tau |P'(i)|} \sum_{p \in P'(i)} \frac{N_1 \cdot f_p + \sum_{p' \in P'(i)} N_2 \cdot f_{p'} + \sum_{n' \in N(i)} N_3 \cdot f_{n'}}{B(i)}, \quad (6)$$

where

$$\begin{cases} M_1 = A(i,p) - exp(f_i^a \cdot f_p^b/\tau), \\ M_2 = exp(-f_i^a \cdot f_{p'}^b/\tau), \\ M_3 = -exp(f_i^a \cdot f_{n'}^b/\tau), \end{cases} \quad \begin{cases} N_1 = B(i), \\ N_2 = -exp(f_i \cdot f_{p'}/\tau), \\ N_3 = -exp(f_i \cdot f_{n'}/\tau). \end{cases} \quad (7)$$

The detailed proof is included in Appendix A. As shown in Eq. 5 and Eq. 7, it is clear that the coefficient $M1, M2 > 0$ and $M3 < 0$, which means the gradient for $f_i^a$ is towards the positive character features $f_p^b$ and $f_{p'}^b$ in base images. $f_{n'}^b$ is considered a negative sample and needs to be distinguished. As shown in Eq. 6 and Eq. 7, conditions are the same for $N1 > 0$ and $N3 < 0$. However, $N2 < 0$ indicates that the positive sample $f_{p'}$ is mistakenly regarded as a negative sample like $f_{n'}$, which is harmful to the learning of clustering characters in the same class.

### 3.3 Training objective

Given base images $\mathbf{x}^b$, the teacher model obtains a sequence of character features $\mathbf{F}^b$ and prediction probabilities $\mathbf{P}^b = \{p_1^b, p_2^b, \cdots, p_T^b\}$. Correspondingly, the student model generates $\mathbf{F}^a$ and $\mathbf{P}^a = \{p_1^a, p_2^a, \cdots, p_T^a\}$ for strong augmented images $\mathbf{x}^a$. Following [54], $\mathbf{P}^a$ is obtained with an additional projector employed in the student model to ensure a stable training process. We also adopt Character-level Consistency Regularization (CCR) to boost the training of our semi-supervised framework. This process can be formulated as

$$\mathcal{L}_{ccr} = \mathbb{I}(S^b > \eta_{ccr}) \frac{1}{T^b} \sum_{t=1}^{T^b} KL(p_t^b, p_t^a), \quad (8)$$

where $\mathbb{I}(\cdot)$ is the indicator function. $T^b$ is the text length of $\mathbf{x}^b$. $\eta_{ccr}$ is the threshold for confidence score $S^b$, which is 1 for labeled data and the cumulative product of the maximum probability sequence

Table 1: Comparison with SOTA methods on common benchmarks and Union-B. * means we use publicly released checkpoints to evaluate the method. † means we reproduce the methods with the same configuration. For training data: SL - MJSynth and SynthText; RL - Real labeled data; RU - Union14M-U; $RU^1$ - Book32, TextVQA, and ST-VQA; $RU^2$ - Places2, OpenImages, and ImageNet ILSVRC 2012. Cur, M-O, Art, Ctl, Sal, M-W, and Gen represent Curve, Multi-Oriented, Artistic, Contextless, Salient, Multi-Words, and General. P(M) means the model size.

| Method | Datasets | Common Benchmarks | | | | | | | Union14M-Benchmark | | | | | | | | P(M) |
|---|---|---|---|---|---|---|---|---|---|---|---|---|---|---|---|---|---|
| | | IIIT | SVT | IC13 | IC15 | SVTP | CUTE | WAVG | Cur | M-O | Art | Con | Sal | M-W | Gen | AVG | |
| CRNN [33] | SL | 82.9 | 81.6 | 91.9 | 69.4 | 70.0 | 65.5 | 78.6 | 7.5 | 0.9 | 20.7 | 25.6 | 13.9 | 25.6 | 32.0 | 18.0 | 8.3 |
| RobustScanner [50] | SL | 95.3 | 88.1 | 94.8 | 77.1 | 79.5 | 90.3 | 88.4 | 43.6 | 7.9 | 41.2 | 42.6 | 44.9 | 46.9 | 39.5 | 38.1 | - |
| SRN [49] | SL | 94.8 | 91.5 | 95.5 | 82.7 | 85.1 | 87.8 | 90.4 | 63.4 | 25.3 | 34.1 | 28.7 | 56.5 | 26.7 | 46.3 | 39.6 | 54.7 |
| ABINet [10] | SL+Wiki | 96.2 | 93.5 | 97.4 | 86.0 | 89.3 | 89.2 | 92.3 | 59.5 | 12.7 | 43.3 | 38.3 | 62.0 | 50.8 | 55.6 | 46.0 | 36.7 |
| VisionLAN [42] | SL | 95.8 | 91.7 | 95.7 | 83.7 | 86.0 | 88.5 | 91.2 | 57.7 | 14.2 | 47.8 | 48.0 | 64.0 | 47.9 | 52.1 | 47.4 | 32.8 |
| MATRN [25] | SL | 96.6 | 95.0 | 97.9 | 86.6 | 90.6 | **93.5** | 93.5 | 63.1 | 13.4 | 43.8 | 41.9 | 66.4 | 53.2 | 57.0 | 48.4 | 44.2 |
| SVTR [9] | SL | 96.0 | 91.5 | 97.1 | 85.2 | 89.9 | 91.7 | 92.3 | 63.0 | 32.1 | 37.9 | 44.2 | 67.5 | 49.1 | 52.8 | 49.5 | 24.6 |
| ParSeq† [2] | SL | 97.0 | 93.6 | 97.0 | 86.5 | 88.9 | 92.3 | 93.3 | 58.2 | 17.2 | 54.2 | 59.4 | 67.7 | 55.8 | 61.1 | 53.4 | 23.8 |
| MGP* [41] | SL | 96.4 | 94.7 | 97.3 | 87.2 | 91.0 | 90.3 | 93.3 | 55.2 | 14.0 | 52.8 | 48.4 | 65.1 | 48.1 | 59.0 | 48.9 | 141.1 |
| CLIP-OCR* [43] | SL | 97.3 | 94.7 | 97.7 | 87.2 | 89.9 | 93.1 | 93.8 | 59.4 | 15.9 | 57.6 | 59.2 | 69.2 | 62.6 | 62.3 | 55.2 | 31.1 |
| LPV* [52] | SL | 97.3 | 94.6 | 97.6 | 87.5 | 90.9 | 94.8 | 94.0 | 68.3 | 21.0 | 59.6 | 65.1 | 76.2 | 63.6 | 62.0 | 59.4 | 35.1 |
| LISTER* [5] | SL | 96.9 | 93.8 | 97.9 | 87.5 | 89.6 | 90.6 | 93.5 | 54.8 | 17.2 | 51.3 | 61.5 | 62.6 | 61.3 | 62.9 | 53.1 | 49.9 |
| CRNN-pr* [1] | RL+$RU^1$ | 90.2 | 86.1 | 91.5 | 77.8 | 74.1 | 81.6 | 85.1 | 32.3 | 2.9 | 40.4 | 52.6 | 21.3 | 38.8 | 49.8 | 34.0 | 8.5 |
| TRBA-pr* [1] | RL+$RU^1$ | 94.9 | 92.4 | 94.2 | 84.8 | 82.8 | 87.9 | 90.7 | 62.9 | 12.5 | 64.7 | **74.2** | 51.2 | **68.1** | 62.0 | 56.5 | 49.9 |
| ABINet* [10] | SL+Wiki+Uber | 96.9 | 94.7 | 97.0 | 85.9 | 89.0 | 89.9 | 93.0 | 60.9 | 14.2 | 47.9 | 46.6 | 65.6 | 49.0 | 57.6 | 48.8 | 36.7 |
| TRBA-cr* [54] | SL+$RU^2$ | 96.5 | **96.3** | 98.3 | 89.3 | **93.3** | 93.4 | 94.5 | **77.5** | 30.4 | 66.0 | 59.8 | 76.3 | 39.6 | 64.6 | 59.2 | 49.9 |
| TRBA-cr† [54] | SL+RU | 95.5 | 94.6 | 97.4 | 87.1 | 90.1 | 91.3 | 92.9 | 70.5 | 23.1 | 62.7 | 54.6 | 71.8 | 36.6 | 63.8 | 54.7 | 49.9 |
| CRNN† | SL | 88.8 | 82.7 | 90.3 | 69.5 | 69.3 | 76.7 | 81.4 | 21.1 | 63.0 | 26.3 | 25.0 | 20.1 | 25.0 | 38.6 | 31.3 | 8.6 |
| CRNN-ViSu | SL+RU | 90.0 | 84.7 | 90.9 | 71.1 | 73.6 | 77.4 | 83.0 | 23.9 | 66.8 | 30.3 | 27.5 | 24.0 | 27.6 | 43.3 | 34.8 | 8.6 |
| TRBA† | SL | 95.6 | 92.7 | 96.3 | 83.4 | 85.6 | 87.9 | 91.2 | 63.8 | 82.1 | 49.1 | 47.4 | 70.2 | 51.8 | 54.8 | 59.9 | 49.9 |
| TRBA-ViSu | SL+RU | 96.4 | 96.1 | 97.2 | 86.2 | 87.8 | 89.9 | 92.9 | 71.8 | 83.9 | 63.4 | 52.3 | 73.2 | 57.5 | 64.4 | 66.6 | 49.9 |
| Baseline | SL | 96.1 | 94.4 | 96.5 | 86.0 | 88.2 | 88.5 | 92.5 | 60.5 | 82.7 | 53.6 | 54.0 | 70.6 | 54.5 | 62.0 | 62.6 | 24.4 |
| ViSu | SL+RU | **97.6** | 96.1 | **98.3** | **89.3** | 91.3 | 92.4 | **94.7** | 71.6 | **85.8** | **66.8** | 57.6 | **80.3** | 62.6 | **71.2** | **70.9** | 24.4 |

$p_t^b$ for unlabeled data. To extract the commonalities of each character and cluster the character features belonging to the same class, we align the character features $\mathbf{F}^b$ and $\mathbf{F}^a$ using CUA Loss, which is described in Sec. 3.2. To equip the model with basic recognition ability, both $\mathbf{x}^b$ and $\mathbf{x}^a$ for labeled data are input into the student model to obtain the predicted sequence $\hat{\mathbf{y}}^b$ and $\hat{\mathbf{y}}^a$. We use character-level cross-entropy loss as the recognition loss, which can be expressed as follows,

$$\mathcal{L}_{rec} = -\frac{1}{T}\sum_{t=1}^{T} log(\hat{\mathbf{y}}_t|\mathbf{y}_t^{gt}), \tag{9}$$

where $\hat{\mathbf{y}}_t$ means the prediction at position $t$ for $\hat{\mathbf{y}}^b$ and $\hat{\mathbf{y}}^a$. $\mathbf{y}_t^{gt}$ is the corresponding label. The overall loss function is

$$\mathcal{L} = \mathcal{L}_{rec} + \mathcal{L}_{ccr} + \lambda\mathcal{L}_{cua}, \tag{10}$$

where $\lambda$ is a hyper-parameter to balance the numeric values and is set to 0.1.

## 4 Experiment

### 4.1 Datasets

Without any laborious annotation process, our method only needs Synthetic Labeled (SL) data and Real Unlabeled (RU) data. SL includes two widely used synthetic datasets MJSynth [14] and SynthText [12], which contain 9M and 7M synthetic images. For real data without annotations, we adopt Union14M-U [15] with a total of 10M refined images from Book32 [13], CC [32], and OpenImages [18]. We evaluate STR models on six common benchmarks, including IIIT [24], SVT [40], IC13 [17], IC15 [16], SVTP [28], and CUTE [30]. Following the previous methods [54, 43], we use IC13-857 and IC15-1811 without images containing non-alphanumeric characters. To further evaluate the performance on difficult text images, we introduce several challenging benchmarks, including Union14M-Benchmark (Union-B) [15], WordArt [45], ArT [6], COCO-Text (COCO) [39], and Uber-Text (Uber) [53]. Union-B comprises Curve, Multi-Oriented, Artistic, Contextless, Salient, Multi-Words, and General with 2,426, 1,369, 900, 779, 1,585, 829, and 400,000 images, respectively.

### 4.2 Evaluation Metrics

The word accuracy is used to evaluate scene text recognition models, which means that all predicted characters in a word must be identical to the label to be considered correct. We further calculate the

Table 2: Comparison with SOTA methods on several challenging benchmarks. All symbols have the same meaning as in Table. 1.

| Method | Datasets | WordArt | ArT | COCO | Uber |
|---|---|---|---|---|---|
| CLIP-OCR [43] | SL | 73.9 | 70.5 | 66.5 | 42.4 |
| CornerTransformer [45] | SL | 70.8 | - | - | - |
| ParSeq [2] | SL | - | 70.7 | 64.0 | 42.0 |
| LISTER* [5] | SL | 69.8 | 70.1 | 65.8 | 49.0 |
| MGP* [41] | SL | 72.4 | 69.0 | 65.4 | 40.7 |
| CRNN-pr* [1] | RL+RU[1] | 53.4 | 58.6 | 56.8 | 45.2 |
| TRBA-pr* [1] | RL+RU[1] | 64.3 | 66.8 | 67.1 | 54.2 |
| ABINet* [10] | SL+Wiki+Uber | 71.3 | 68.3 | 63.1 | 39.6 |
| TRBA-cr* [54] | SL+RU[2] | 80.2 | 73.3 | 70.3 | 39.3 |
| TRBA-cr† [54] | SL+RU | 76.7 | 72.0 | 69.3 | 40.0 |
| CRNN† | SL | 54.0 | 58.8 | 46.2 | 42.4 |
| CRNN-ViSu | SL+RU | 56.3 | 61.7 | 47.1 | 45.9 |
| TRBA† | SL | 69.5 | 69.9 | 63.3 | 54.7 |
| TRBA-ViSu | SL+RU | 77.7 | 72.8 | 69.8 | 58.4 |
| Baseline | SL | 72.3 | 74.7 | 67.6 | 60.3 |
| Baseline-ViSu | SL+RU | **80.5** | **78.8** | **74.9** | **67.2** |

weighted average score (WAVG) for six common benchmarks. Because the number of General in Union-B is much larger than the others, we report their average score (AVG).

## 4.3 Comparison with SOTA

In Table. 1, we compare our method with previous state-of-the-art methods on common benchmarks and Union-B. Table. 2 displays the performance on several challenging datasets. Our method achieves higher accuracy with smaller parameters on all benchmarks than fully-supervised methods. Compared with the semi-supervised methods, because the samples in common benchmarks are relatively easy, our method has an accuracy of 0.2% higher than TRBA-cr [54]. When evaluated on challenging benchmarks, our method shows significant performance superiority, namely 11.7%, 0.3%, 5.5%, 4.6%, and 27.9% accuracy increase on Union-B, WordArt, ArT, COCO, and Uber, respectively. Because the released checkpoint of TRBA-cr [54] uses the different real unlabeled datasets that are not publicly available, for a fair comparison, we reproduce the official code and train with the same real unlabeled datasets as ours. Benefiting from Unified Representation Forms (URF), our method surpasses them by a large margin on the benchmarks containing enormous multi-directional texts (85.8% vs 23.1% on Multi-Oriented and 67.2% vs 40.0% on Uber). For other benchmarks with challenging texts, benefiting from OGS and CUA Loss, our method shows significant improvement in Artistic and WordArt.

We further apply our proposed framework to other models. As shown in the last six lines in Table. 1 and Table. 2, to make the model robust to multi-directional text, the methods without ViSu and Baseline also adopt URF. Therefore, they perform way better than other supervised and semi-supervised methods on Multi-Oriented and Uber. Equipped with ViSu, all the listed models gain at least 1.6% on common benchmarks, 3.5% on Union-B, 2.3% on WordArt, 2.9% on ArT, 0.9% on COCO, and 3.5% on Uber, proving the effectiveness of our proposed framework.

## 4.4 Ablation Study

### 4.4.1 Online Generation Strategy

We generate 1.6M samples using OGS to replace MJSynth [14] and SynthText [12] datasets. Because MJSynth and SynthText contain approximately 16M images in total, we utilize only 10% of them for fair comparison. As shown in Table. 3, ViSu trained with 10% synthetic data achieves an average accuracy of 58.2%. Upon incorporating OGS and employing the corresponding losses, we observe an improvement of 2.1%, underscoring the effectiveness of our proposed OGS and CUA

Table 3: Ablation experiments with different training data.

| Method | Datasets | Cur | M-O | Art | Con | Sal | M-W | Gen | AVG |
|---|---|---|---|---|---|---|---|---|---|
| ParSeq [2] | 10% (MJ + ST) + OGS | 54.3 | 15.7 | 52.3 | 53.2 | 67.3 | 55.9 | 58.8 | 51.1 |
| MGP [41] | 10% (MJ + ST) + OGS | 46.8 | 10.5 | 49.9 | 33.0 | 55.1 | 26.7 | 55.8 | 39.7 |
| CLIPOCR [43] | 10% (MJ + ST) + OGS | 57.1 | 13.0 | 57.1 | 49.2 | 65.5 | 60.2 | 59.8 | 51.7 |
| LPV [52] | 10% (MJ + ST) + OGS | 58.6 | 12.9 | 53.3 | 53.3 | 67.4 | 59.3 | 56.9 | 51.7 |
| LISTER [5] | 10% (MJ + ST) + OGS | 52.0 | 13.7 | 48.9 | 54.4 | 59.8 | 54.5 | 61.0 | 49.2 |
| TRBA-cr [54] | 10% (MJ + ST) + OGS | 67.1 | 17.4 | 58.6 | 51.1 | 67.7 | 33.5 | 57.4 | 50.4 |
| ViSu | 10% (MJ + ST) | 57.5 | 79.6 | 49.8 | 44.4 | 66.8 | 50.4 | 59.2 | 58.2 |
| ViSu | OGS | 1.7 | 25.1 | 5.8 | 4.9 | 3.0 | 6.9 | 10.2 | 8.2 |
| ViSu | 10% (MJ + ST) + OGS | 60.8 | 80.8 | 52.4 | 47.1 | 69.1 | 51.8 | 59.9 | **60.3** |

Table 4: Ablation experiments with different configurations. URF means unified representation forms. OGS represents the online generation strategy. RU indicates whether to use real unlabeled data. $CC^1$ means only aligning with other characters in strong augmented images. $CC^2$ means the alignment between strong augmented images and base images.

| Consistency loss | Alignment loss | URF | RU | OGS | Cur | M-O | Art | Con | Sal | M-W | Gen | AVG |
|---|---|---|---|---|---|---|---|---|---|---|---|---|
| CE | - | - | - | - | 58.0 | 17.0 | 52.9 | 55.4 | 64.9 | 54.8 | 61.3 | 52.0 |
| CE | CUA Loss | ✓ | ✓ | ✓ | 71.2 | 86.2 | 65.8 | 58.5 | 79.4 | 63.4 | 70.9 | 70.8 |
| KL-div | - | ✓ | ✓ | ✓ | 69.9 | 83.9 | 64.1 | 54.9 | 76.0 | 60.8 | 67.6 | 68.2 |
| KL-div | $CC^1$ Loss | ✓ | ✓ | ✓ | 69.4 | 84.7 | 64.8 | 56.5 | 76.1 | 62.2 | 69.8 | 69.1 |
| KL-div | $CC^2$ Loss | ✓ | ✓ | ✓ | 70.6 | 84.7 | 65.4 | 58.8 | 79.4 | 61.3 | 70.6 | 70.1 |
| KL-div | CUA Loss | - | ✓ | ✓ | 65.9 | 34.6 | 64.6 | 60.7 | 75.2 | 62.7 | 66.2 | 61.4 |
| KL-div | CUA Loss | ✓ | - | ✓ | 64.4 | 83.9 | 56.0 | 55.5 | 73.2 | 55.9 | 62.6 | 64.5 |
| KL-div | CUA Loss | ✓ | ✓ | - | 68.2 | 84.5 | 62.0 | 58.5 | 75.8 | 62.5 | 69.7 | 68.8 |
| KL-div | CUA Loss | ✓ | ✓ | ✓ | 71.6 | 85.8 | 66.8 | 57.6 | 80.3 | 62.6 | 71.2 | 70.9 |

Loss. However, the model trained solely on OGS samples struggles to recognize texts. We attribute the poor performance to the diverse styles of characters, domain gap with Union-B, and varied orientations of text images generated by OGS. Models trained from scratch with such challenging data struggle to extract practical character information and are susceptible to background noise, hindering the acquisition of basic text recognition ability. By incorporating simple synthetic data to establish a foundation for recognition abilities and boosting it with our proposed OGS and CUA Loss targeted for challenging texts, the model is able to achieve promising performance. Furthermore, we add OGS samples to the training data of other methods to ensure the consistency of the dataset. The experimental results strongly demonstrate that the superiority does not come from the use of additional data but from the promotion of the semi-supervised learning framework by OGS.

### 4.4.2 Discussion about loss functions

We evaluate the performance of the model when using cross entropy and KL-divergence as consistency loss, respectively. As shown in Table. 4, they have similar performance. For the alignment loss, both CC Loss and CUA Loss can bring performance improvement. However, as stated in Sec. 3.2, the original CC Loss only clusters the character features in the minibatch, referred to as $CC^1$. Adopting our framework and aligning the character features in strong augmented images to the base images ($CC^2$) achieve 1.0% higher performance. Moreover, because some positive samples are mistakenly treated as negatives, adopting our CUA Loss brings an accuracy gain of 0.8%, demonstrating the effectiveness of our CUA Loss.

### 4.4.3 Ablation on training setting

As shown in Table. 4, because Curve and Multi-Oriented have many samples with multiple reading orders and character orientations, the model without URF has significantly lower accuracy on these benchmarks. When the model is not trained with real unlabeled data, it only reaches 64.5% accuracy. Compared to the baseline in Table. 1, the proposed OGS, along with the consistency and alignment losses, brings a 1.9% accuracy improvement, which proves that our OGS and CUA Loss improve the

recognition performance of the model on challenging text images when trained using only simple synthetic data. The experiments in the last two lines in Table. 4 also prove the superiority of OGS.

## 5 Conclusion

In this paper, we attribute the bottleneck of STR models to the insufficient exploration of character morphologies. Therefore, considering the human cognitive process, we develop a semi-supervised framework by viewing and summarizing. In the viewing process, we first adopt the Mean-Teacher framework to introduce real unlabeled data. Secondly, an Online Generation Strategy is proposed to explore the potential of synthetic data and raise the performance upper bound of the semi-supervised framework. In the summarizing process, a novel Character Unidirectional Alignment Loss is proposed to theoretically correct the previous formula error of mistakenly treating some positive samples with negative ones. Extensive experiments show that our method achieves state-of-the-art performance on all benchmarks, with particularly significant superiority on the challenging Union-B.

## Acknowledgments

This work is supported by the National Key Research and Development Program of China (2022YFB3104700), the National Nature Science Foundation of China (U23B2028, 62121002, 62102384). We acknowledge the support of GPU cluster built by MCC Lab of Information Science and Technology Institution, USTC. We also thank the USTC supercomputing center for providing computational resources for this project.

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

# A    Grandient of Loss Functions

For Character Unidirectional Alignment (CUA) Loss, we have

$$\mathcal{L}_{cua} = -\sum_{i \in I} \frac{1}{|P(i)|} \sum_{p \in P(i)} log \frac{exp(f_i^a \cdot f_p^b/\tau)}{A(i,p)}, \tag{11}$$

$$A(i,p) = exp(f_i^a \cdot f_p^b/\tau) + \sum_{p' \in P'(p)} exp(-f_i^a \cdot f_{p'}^b/\tau) + \sum_{n' \in N(i)} exp(f_i^a \cdot f_{n'}^b/\tau). \tag{12}$$

The derivation of the gradient for CUA loss:

$$\begin{aligned}
\frac{\partial A(i,p)}{\partial f_i^a} = &\frac{1}{\tau}[exp(f_i^a \cdot f_p^b/\tau)f_p^b - \sum_{p' \in P'(p)} exp(-f_i^a \cdot f_{p'}^b/\tau)f_{p'}^b \\
&+ \sum_{n' \in N(i)} exp(f_i^a \cdot f_{n'}^b/\tau)f_{n'}^b],
\end{aligned} \tag{13}$$

$$\begin{aligned}
\frac{\partial \mathcal{L}_{cua}}{\partial f_i^a} &= -\frac{1}{|P(i)|} \sum_{p \in P(i)} \frac{\partial(f_i^a \cdot f_p^b/\tau)}{\partial f_i^a} - \frac{\partial log(A(i,p))}{\partial f_i^a} \\
&= -\frac{1}{|P(i)|} \sum_{p \in P(i)} \frac{f_p^b}{\tau} - \frac{\partial log(A(i,p))}{\partial A(i,p)} \frac{\partial A(i,p)}{\partial f_i^a} \\
&= -\frac{1}{\tau|P(i)|} \sum_{p \in P(i)} \frac{M_1 \cdot f_p^b + \sum_{p' \in P'(p)} M_2 \cdot f_{p'}^b + \sum_{n' \in N(i)} M_3 \cdot f_{n'}^b}{A(i,p)},
\end{aligned} \tag{14}$$

where

$$\begin{cases}
M_1 = A(i,p) - exp(f_i^a \cdot f_p^b/\tau), \\
M_2 = exp(-f_i^a \cdot f_{p'}^b/\tau), \\
M_3 = -exp(f_i^a \cdot f_{n'}^b/\tau).
\end{cases} \tag{15}$$

And the Character Contrastive (CC) Loss [45] can be expressed as

$$\mathcal{L}_{cc} = -\sum_{i \in I} \frac{1}{|P'(i)|} \sum_{p \in P'(i)} log \frac{exp(f_i \cdot f_p/\tau)}{B(i)}, \tag{16}$$

$$B(i) = \sum_{p' \in P'(i)} exp(f_i \cdot f_{p'}/\tau) + \sum_{n' \in N(i)} exp(f_i \cdot f_{n'}/\tau), \tag{17}$$

The derivation of the gradient for CC loss:

$$\frac{\partial B(i)}{\partial f_i} = \frac{1}{\tau}[\sum_{p' \in P'(i)} exp(f_i \cdot f_{p'}/\tau)f_{p'} + \sum_{n' \in N(i)} exp(f_i \cdot f_{n'}/\tau)f_{n'}], \tag{18}$$

$$\begin{aligned}
\frac{\partial \mathcal{L}_{cc}}{\partial f_i} &= -\frac{1}{|P'(i)|} \sum_{p \in P'(i)} \frac{\partial(f_i \cdot f_p/\tau)}{\partial f_i} - \frac{\partial log((B(i))}{\partial f_i} \\
&= -\frac{1}{|P'(i)|} \sum_{p \in P'(i)} \frac{f_p}{\tau} - \frac{\partial log(B(i))}{\partial B(i)} \frac{\partial B(i)}{\partial f_i} \\
&= -\frac{1}{\tau|P'(i)|} \sum_{p \in P'(i)} \frac{N_1 \cdot f_p + \sum_{p' \in P'(i)} N_2 \cdot f_{p'} + \sum_{n' \in N(i)} N_3 \cdot f_{n'}}{B(i)},
\end{aligned} \tag{19}$$

where

$$\begin{cases} N_1 = B(i), \\ N_2 = -exp(f_i \cdot f_{p'}/\tau), \\ N_3 = -exp(f_i \cdot f_{n'}/\tau). \end{cases} \tag{20}$$

# B  Experiment

## B.1  Implementation Details

In the model configuration, our model consists of a transformer-based encoder with 12 transformer blocks and 6 attention heads and a transformer-based decoder with 1 transformer block and 12 attention heads. All images are resized to $100 \times 32$, and the patch size is $8 \times 4$. The maximum length T is set to 25. The character set size is 36, including 10 digits and 26 alphabets.

For training settings, the network is trained in an end-to-end manner without pre-training. We adopt AdamW optimizer and one-cycle [35] learning rate scheduler with a maximum learning rate of 6e-4. The batchsize is 384 for both synthetic data and real unlabeled data. We set the EMA smoothing factor $\alpha = 0.999$, aspect ratio thresh $r = 1.3$, confidence threshold $\eta_{ccr} = 0.5$, $\eta_{cua} = 0.7$, and temperature factor $\tau = 0.1$. Due to the Online Generation Strategy (OGS) and Unified Representation Forms (URF), all images are rotated 180 degrees with a probability of 0.5 during the training stage. Following [2], the student model takes strong augmented images for labeled and unlabeled data. The unlabeled real images fed into the teacher model are weakly augmented, where only color jitter is used, including brightness, contrast, saturation, and hue. ViSu is trained on 4 NVIDIA RTX 4090 GPUs.

Table 5: Comparison with state-of-the-art methods on common benchmarks and Union-B. * means we use publicly released checkpoints to evaluate the model. † means we reproduce the methods with the same configuration. For training data: RL - Union14M-L; $RL^1$ - Combination of 11 real labeled datasets; RU - Union14M-U; $RU^1$ - Book32, TextVQA, and ST-VQA. Cur, M-O, Art, Ctl, Sal, M-W, and Gen represent Curve, Multi-Oriented, Artistic, Contextless, Salient, Multi-Words, and General. P(M) means the model size.

| Method | Datasets | Common Benchmarks | | | | | | | Union14M-Benchmarks | | | | | | | | P(M) |
|---|---|---|---|---|---|---|---|---|---|---|---|---|---|---|---|---|---|
| | | IIIT | SVT | IC13 | IC15 | SVTP | CUTE | WAVG | Cur | M-O | Art | Con | Sal | M-W | Gen | AVG | |
| CRNN [33] | RL | 90.8 | 83.8 | 91.8 | 71.8 | 70.4 | 80.9 | 83.1 | 19.4 | 4.5 | 34.2 | 44.0 | 16.7 | 35.7 | 60.4 | 30.7 | 8.3 |
| RobustScanner [50] | RL | 96.8 | 92.4 | 95.7 | 86.4 | 83.9 | 93.8 | 92.3 | 66.2 | 54.2 | 61.4 | 72.7 | 60.1 | 74.2 | 75.7 | 66.4 | - |
| SRN [49] | RL | 95.5 | 89.5 | 94.7 | 79.1 | 83.9 | 91.3 | 89.3 | 49.7 | 20.0 | 50.7 | 61.0 | 43.9 | 51.5 | 62.7 | 48.5 | 54.7 |
| ABINet [10] | RL | 97.2 | 95.7 | 97.2 | 87.6 | 92.1 | 94.4 | 93.9 | 75.0 | 61.5 | 65.3 | 71.1 | 72.9 | 59.1 | 79.4 | 69.2 | 36.7 |
| VisionLAN [42] | RL | 96.3 | 91.3 | 95.1 | 83.6 | 85.4 | 92.4 | 91.2 | 70.7 | 57.2 | 56.7 | 63.8 | 67.6 | 47.3 | 74.2 | 62.5 | 32.8 |
| MATRN [25] | RL | 98.2 | 96.9 | 97.9 | 88.2 | 94.1 | 97.9 | 95.0 | 80.5 | 64.7 | 71.1 | 74.8 | 79.4 | 67.6 | 77.9 | 74.6 | 44.2 |
| SVTR [9] | RL | 95.9 | 92.4 | 95.5 | 83.9 | 85.7 | 93.1 | 91.3 | 72.4 | 68.2 | 54.1 | 68.0 | 71.4 | 67.7 | 77.0 | 68.4 | 24.6 |
| ParSeq† [2] | RL | 98.5 | 97.7 | 98.0 | 88.7 | 94.7 | 96.5 | 95.3 | 84.3 | 86.0 | 76.2 | 80.7 | 82.5 | 82.5 | 84.1 | 82.3 | 23.8 |
| MGP† [41] | RL | 97.2 | 97.7 | 96.8 | 87.2 | 93.6 | 94.8 | 94.1 | 78.8 | 74.3 | 67.7 | 68.7 | 75.7 | 60.0 | 80.1 | 72.2 | 141.1 |
| LPV† [52] | RL | 98.7 | **98.6** | 97.9 | 88.7 | 93.9 | 96.2 | 95.4 | 85.2 | 75.9 | 74.8 | 80.5 | 83.3 | 82.2 | 82.8 | 80.7 | 35.1 |
| LISTER† [5] | RL | 98.1 | 97.7 | 97.2 | 86.3 | 91.8 | 94.8 | 94.1 | 70.9 | 51.1 | 65.4 | 73.3 | 66.9 | 77.6 | 77.9 | 69.0 | 49.9 |
| CLIP-OCR† [43] | RL | **98.8** | 98.3 | 98.2 | 88.8 | 94.3 | 97.6 | 95.5 | 84.6 | 83.1 | 76.3 | 80.0 | 81.3 | 81.8 | 83.9 | 81.6 | 31.1 |
| CRNN-pr* [1] | $RL^1$+$RU^1$ | 90.2 | 86.1 | 91.5 | 77.8 | 74.1 | 81.6 | 85.1 | 32.3 | 2.9 | 40.4 | 52.6 | 21.3 | 38.8 | 49.8 | 34.0 | 8.5 |
| TRBA-pr* [1] | $RL^1$+$RU^1$ | 94.9 | 92.4 | 94.2 | 84.8 | 82.8 | 87.9 | 90.7 | 62.9 | 12.5 | 64.7 | 74.2 | 51.2 | 68.1 | 62.0 | 56.5 | 49.9 |
| TRBA-cr† [54] | RL+RU | 98.5 | 98.3 | 97.7 | 89.8 | 94.3 | 96.5 | 95.6 | 83.4 | 81.7 | 74.8 | 78.9 | 83.6 | 79.1 | 81.9 | 80.5 | 49.9 |
| MAERec-S [15] | RL+RU | 98.0 | 96.8 | 97.6 | 87.1 | 93.2 | 97.9 | 94.5 | 81.4 | 71.4 | 72.0 | 82.0 | 78.5 | 82.4 | 82.5 | 78.6 | 34.0 |
| MAERec-B [15] | RL+RU | 98.5 | 97.8 | 98.1 | 89.5 | 94.4 | **98.6** | 95.6 | 88.8 | 83.9 | **80.0** | **85.5** | 84.9 | **87.5** | **85.8** | 85.2 | 135.5 |
| Baseline | RL | 98.6 | 98.0 | **98.3** | 89.4 | 94.3 | 96.5 | 95.7 | 88.8 | 95.8 | 78.1 | 83.4 | 86.3 | 83.6 | 82.7 | 85.5 | 24.4 |
| ViSu | RL+RU | 98.5 | 98.3 | 97.8 | **90.4** | **96.3** | 97.6 | **96.0** | **90.7** | **96.1** | 79.4 | 85.4 | **87.1** | 86.3 | 83.1 | **86.9** | 24.4 |

## B.2  Training with Real Labeled Data

Our method aims to boost the scene text recognition (STR) model without any human costs. However, to demonstrate the effectiveness of our method when training with real labeled data, we replace the synthetic labeled data with Union14M-L [15], which consists of 3.2M refined images from 14 publicly available datasets. For the convenience of comparison and following [15], we employ IC13-1015 and IC15-2077 to evaluate all the STR methods.

As shown in Table. 5 and Table. 6, our method achieves higher accuracy than all fully supervised methods. Compared to the state-of-the-art pretraining method MAERec [15], with approximately 1/5

Table 6: Comparison with state-of-the-art methods on several challenging benchmarks. All symbols have the same meaning as in Table. 5.

| Method | Datasets | WordArt | ArT | COCO | Uber |
|---|---|---|---|---|---|
| ParSeq† [2] | RL | 83.1 | 83.1 | 77.8 | 87.4 |
| MGP† [41] | RL | 80.5 | 79.6 | 74.1 | 80.7 |
| LPV† [52] | RL | 82.5 | 82.4 | 76.8 | 81.8 |
| LISTER† [5] | RL | 74.8 | 78.3 | 72.7 | 81.3 |
| CLIP-OCR† [43] | RL | 83.5 | 82.7 | 78.3 | 86.3 |
| CRNN-pr* [1] | $RL^1+RU^1$ | 53.4 | 58.6 | 56.8 | 45.2 |
| TRBA-pr* [1] | $RL^1+RU^1$ | 64.3 | 66.8 | 67.1 | 54.2 |
| TRBA-cr† [54] | RL+RU | 82.7 | 81.9 | 77.6 | 77.7 |
| ViSu | RL+RU | **85.9** | **84.9** | **80.4** | **91.6** |

Table 7: Recognition efficiency of different methods. WAVG means weighted average accuracy on six common benchmarks. AVG means average accuracy on Union-B. All methods are trained with real data.

| Methods | Speed(ms/img) | Params(M) | WAVG | AVG |
|---|---|---|---|---|
| ABINet [10] | 17.3 | 36.7 | 93.9 | 69.2 |
| ParSeq [2] | 12.0 | 23.8 | 95.3 | 82.3 |
| MGP [41] | 9.0 | 141.1 | 94.1 | 72.2 |
| LPV [52] | 20.0 | 35.1 | 95.4 | 80.7 |
| LISTER [5] | 24.9 | 49.9 | 94.1 | 69.0 |
| CLIP-OCR [43] | 20.2 | 31.1 | 95.5 | 81.6 |
| TRBA-cr [54] | 15.3 | 49.9 | 95.6 | 80.5 |
| MAERec-S [15] | 282.2 | 34.0 | 94.5 | 78.6 |
| MAERec-B [15] | 308.1 | 135.5 | 95.6 | 85.2 |
| ViSu | 17.7 | 24.4 | 96.0 | 86.9 |

of the parameters, our method outperforms it by 0.4% on common benchmarks and by 1.7% on Union-B [15]. Furthermore, for a fair comparison, we reproduce TRBA-cr [54] using the same datasets as ours. The experimental results show that ViSu performs better on all benchmarks, especially on challenging benchmarks (86.9% vs 80.5% on Union-B [15], 85.9% vs 82.7% on WordArt [45], 84.9% vs 81.9% on ArT [6], 80.4% vs 77.6% on COCO [39], and 91.6% vs 77.7% on Uber [53]). We attribute this superior performance to the increased diversity in character morphology from the Online Generation Strategy and the clustering of similar character features facilitated by the CUA Loss.

## B.3 Efficiency

Table. 7 presents the efficiency metrics. Speed is calculated by averaging the inference time over 4,000 images. All experiments are conducted on an NVIDIA RTX 4090 GPU with a batch size of 1. The reported methods are trained using real data. In comparison to MAERec-B [15], our method achieves higher accuracy (86.9% vs 85.2%) with a significantly lighter model, *i.e.*, approximately 1/5 parameter size and 1/17 inference time. Compared to models of similar parameter sizes and inference times, ViSu outperforms them by at least 4.6%, demonstrating the effectiveness of our approach.

## B.4 Discuss about configuration settings for OGS

Table. 8 shows the performance with different configuration of OGS. The first 2 to 5 rows indicate that applying independent random font and unified random character orientation to each character in

Table 8: Performance on Union-B with different configuration settings for OGS. The first row represents the baseline model without OGS. Character-level means that all characters in a sample have independent random font or orientation. Instance-level means that means that all characters in a sample have a unified font or orientation, but different samples are independent.

| random font | random orientation | background | text color | Cur | M-O | Art | Con | Sal | M-W | Gen | AVG |
|---|---|---|---|---|---|---|---|---|---|---|---|
| - | - | - | - | 68.2 | 84.5 | 62.0 | 58.5 | 75.8 | 62.5 | 69.7 | 68.8 |
| character-level | instance-level | - | - | 71.6 | 85.8 | 66.8 | 57.6 | 80.3 | 62.6 | 71.2 | **70.9** |
| instance-level | instance-level | - | - | 71.1 | 85.1 | 67.2 | 56.7 | 79.6 | 63.4 | 71.7 | 70.7 |
| character-level | character-level | - | - | 71.1 | 85.1 | 66.7 | 56.4 | 80.8 | 63.1 | 70.9 | 70.6 |
| instance-level | character-level | - | - | 70.3 | 86.4 | 65.4 | 56.9 | 79.9 | 62.5 | 70.4 | 70.3 |
| character-level | instance-level | ✓ | - | 70.3 | 84.3 | 64.9 | 57.5 | 78.7 | 61.7 | 68.8 | 69.5 |
| character-level | instance-level | - | ✓ | 71.2 | 85.3 | 66.3 | 58.3 | 79.4 | 63.2 | 70.9 | 70.7 |

a sample leads to better performance. This is because random fonts have more diversity in character glyphs, making the model robust to character morphology. All characters in real scene text images tend to have the same orientation, thus adopting a uniform orientation in one sample is a better solution. Moreover, adding a random background and a random color to the text both bring a slight performance degradation. Because OGS samples act as base images in the teacher model to guide the character feature alignment in the student model, they are expected to have noise-free features to enhance the robustness of the model to character morphology. Therefore, introducing extra backgrounds and colors is redundant. In general, changing the configuration settings of OGS brings some slight performance fluctuations, but all are better than not using OGS, which proves the effectiveness of our approach.

## B.5 The influence of hyper-parameters in CUA Loss

We conduct several experiments to demonstrate the influence of $\lambda$, $\eta_{cua}$, and $\tau$. $\lambda$ is the weight of CUA Loss. When $\lambda$ is set to 0, indicating that the model is trained without CUA Loss, the accuracy can only reach 68.2%. A small $\lambda$ implies minimal clustering of characters into the same class and makes the model focus more on the basic recognition loss. The confidence threshold $\eta_{cua}$ determines the characters involved in the CUA Loss. A low threshold introduces many uncertain characters, where incorrect predictions can harm the alignment. Conversely, a high threshold reduces the number of characters participating in the alignment and clustering, similar to a small $\lambda$. The temperature factor $\tau$ adjusts the smoothness of character features. A small $\tau$ focuses more on challenging characters but sharpens the feature distribution. As shown in Table. 9, changes in these parameters do not significantly affect model performance. To achieve relatively better performance, we set $\lambda = 0.1$, $\eta_{cua} = 0.7$, and $\tau = 0.1$ as our default configuration.

## B.6 Qualitative Analysis

Fig. 4(a) displays several challenging examples. The baseline can correctly recognize simple images with multiple directions, benefiting from URF. However, it fails to recognize complex characters due to a lack of adaptation to character morphology. The supervised method ParSeq and semi-supervised method TRBA-cr have difficulty recognizing these challenging text images. Fig. 4(b) and (c) show the distribution of character features from t-SNE [38]. Compared with CC Loss, CUA Loss better clusters characters in the same class and makes character features in different classes more distinguishable.

## B.7 Limitations

Our method has difficulty recognizing scene text images with extreme aspect ratios. As shown in Fig. 5, when the text in a scene image is very long, it usually has a large aspect ratio. However, all images are resized to a fixed size before being fed into the text recognizer, which results in drastic changes in the aspect ratio of long text images. The information loss caused by resizing the images impairs the extraction of character features. This is a common problem in scene text recognition task and can be alleviated by splitting long texts into multiple segments for recognition and then concatenating them together.

Table 9: Performance on Union-B with different hyper-parameters in CUA Loss. The model is trained on synthetic datasets and Union14-U.

| Hyperparameters | AVG |
|---|---|
| $\lambda = 0$ | 68.2 |
| $\lambda = 0.01, \eta_{cua} = 0.7, \tau = 0.1$ | 69.0 |
| $\lambda = 0.1, \eta_{cua} = 0.7, \tau = 0.1$ | **70.9** |
| $\lambda = 1, \eta_{cua} = 0.7, \tau = 0.1$ | 68.6 |
| $\lambda = 0.1, \eta_{cua} = 0.5, \tau = 0.1$ | 70.5 |
| $\lambda = 0.1, \eta_{cua} = 0.9, \tau = 0.1$ | 70.6 |
| $\lambda = 0.1, \eta_{cua} = 0.7, \tau = 0.05$ | 70.6 |
| $\lambda = 0.1, \eta_{cua} = 0.7, \tau = 0.2$ | 70.3 |

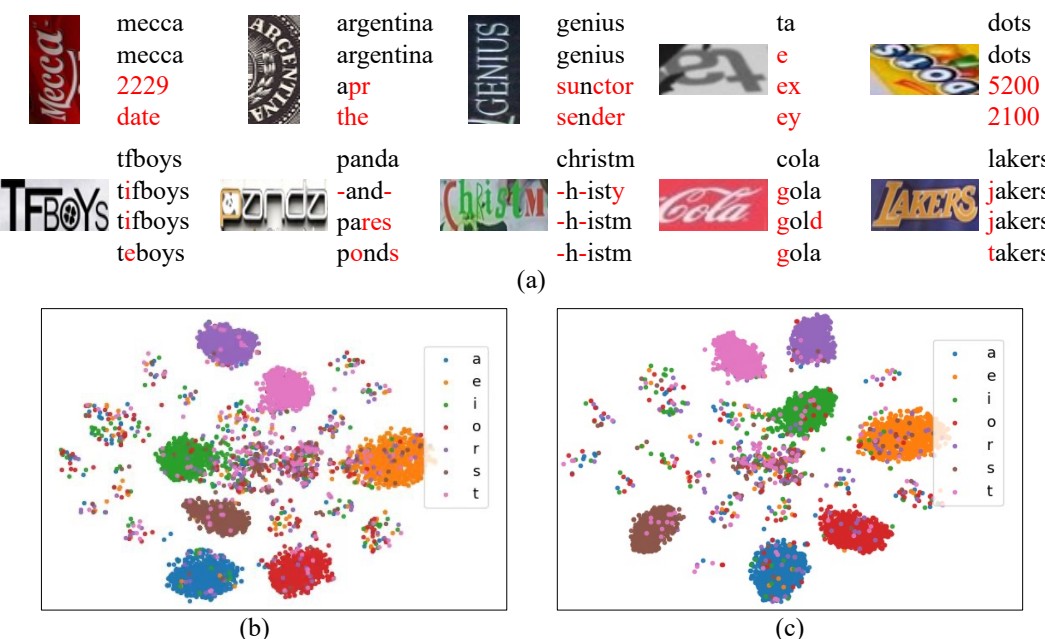

(a)

(b)      (c)

Figure 4: (a) shows several challenging examples. The four lines from top to bottom represent the recognition results from ViSu, Baseline, ParSeq, and TRBA-cr. The first row shows examples with multiple directions. The second row displays examples with artistic or distorted characters. (b) and (c) are the visualizations of character features for CC Loss and CUA Loss, respectively.

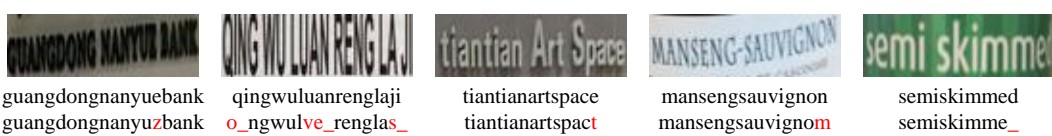

Figure 5: Failure cases. The first line is the ground-truth, and the second line is the recognition results.

