# OpenReview forum: "Boosting Semi-Supervised Scene Text Recognition via Viewing and Summarizing"
_NeurIPS.cc/2024/Conference — NeurIPS 2024 poster_

### Official Review · Reviewer_qi8n · 2024-06-30

**Soundness:** 3
**Presentation:** 3
**Contribution:** 3
**Rating:** 7
**Confidence:** 5

**Summary:**

This paper proposes a ViSu framework to enhance text recognition by viewing and summarizing. For the viewing process, the authors generate various text images to lead the model to focus on different text styles. For the summarizing process, they analyze the drawback in existing character consistency loss and propose a CUA loss to cluster the character under different styles. As a result, the proposed framework outperforms other existing methods on multiple datasets.

**Strengths:**

1. The design of OGS can obtain promising improvements without additional human cost.
2. The authors point out that previous consistency loss mistakenly regards some positive samples as negative samples, which is an important theoretical drawback in existing metric learning methods. And the proposed CUA loss provides a solution for this issue.
3. The proposed method gets SOTA performance on both common and challenging datasets.
4. The proposed framework can be applied on multiple text recognition models seamlessly.

**Weaknesses:**

1. To further verify the effectiveness of OGS. Authors should compare the performance of replacing OGS with normal synthetic data.
2. Tab.5 lacks the comparison with the baseline model.
3. In Tab.5, it seems that MAERec-B performs better on the Union-14m dataset.

**Questions:**

1. As the semi-supervised framework can use large-scale unlabeled real data for training, why does ViSu require OGS to generate synthetic data?
2. To verify the fairness, do other semi-supervised methods in Tab. 1 and 5 have the same implementation details as ViSu?

**Limitations:**

Please refer to Weaknesses.

---

> ### Author Rebuttal · Authors · 2024-08-03
>
> Many thanks for the feedback. In the following we address the weaknesses and questions pointed out by the reviewer.
>
> > Q1: To further verify the effectiveness of OGS. Authors should compare the performance of replacing OGS with normal synthetic data.
>
> A1: Replacing OGS with normal synthetic data means adding a background to the synthetic samples, incorporating color to the text, and using the same fonts and orientations for the characters within each sample. We have validated these elements separately in Appendix. B.4. Real scene text images often have characters with consistent orientations, thus generating characters with the same orientation in one sample is advantageous. Because OGS samples serve as base images and aim to enrich character morphologies without noises, introducing extra backgrounds and colors is redundant. However, applying randomized fonts to each character is shown to be effective for enhancing the robustness of the model.
>
> > Q2: Tab.5 lacks the comparison with the baseline model.
>
> A2: Thanks for pointing this out. Our method aims to elevate the performance upper bound of model recognition without incurring additional human costs, relying on simple synthetic labeled data and real unlabeled data. Table. 5 is to demonstrate that our method still performs well when introducing manual annotations, so there is no obvious definition of baseline. We show the results of training with only real labeled data below. From the experimental metrics, we can see that our ViSu can still yield further improvement with the introduction of manual annotations.
>
> | Method | Datasets | IIIT | SVT | IC13 | IC15 | SVTP | CUTE | WAVG | Cur | M-O | Art | Con | Sal | M-W | Gen | AVG
> |  ----  | ----  | ----  | ----  | ----  | ----  | ----  | ----  | ----  | ----  | ----  | ----  | ----  | ----  | ----  | ----  | ----
> | Baseline | RL | **98.6** | 98.0 | **98.3** | 89.4 | 94.3 | 96.5 | 95.7 | 88.8 | 95.8 | 78.1 | 83.4 | 86.3 | 83.6 | 82.7 | 85.5
> | ViSu | RL+RU | 98.5 | **98.3** | 97.8 | **90.4** | **96.3** | **97.6** | **96.0** | **90.7** | **96.1** | **79.4** | **85.4** | **87.1** | **86.3** | **83.1** | **86.9**
>
>
> > Q3: In Tab.5, it seems that MAERec-B performs better on the Union-14m dataset.
>
> A3: MAERec-B performs better accuracy than ours on the some benchmarks of Union14M-Benchmark. However, our method achieves 1.7% higher average accuracy. It is worth noting that, as shown in Table. 7, MAERec-B has approximately 5 times the number of model parameters and about 17 times the inference time. Our ability to achieve higher accuracy with a much lighter model underscores the effectiveness of our method. Compared to MAERec-S, which has 1.4 times the number of parameters, the inference time is about 16 times longer than ours, and the accuracy is significantly lower than ours (78.6% vs 86.9%).
>
> > Q4: As the semi-supervised framework can use large-scale unlabeled real data for training, why does ViSu require OGS to generate synthetic data?
>
> A4: Both real unlabeled data and labeled data are essential for a semi-supervised framework. Semi-supervised scene text recognizers require annotated data to provide the model with basic character recognition capabilities. Our OGS is designed to enrich the character morphology and maximize the potential of simple data. Therefore, using both real unlabeled data and OGS samples simultaneously is complementary, as both aim to improve the performance from different perspectives.
>
> > Q5: To verify the fairness, do other semi-supervised methods in Tab. 1 and 5 have the same implementation details as ViSu?
>
> A5: The methods marked with * in Table. 1 and Table. 5 mean that we evaluate the officially released checkpoints. The absence of any mark means the metrics reported in the paper. The above methods follow the same evaluation metrics as ours. For the methods marked with $\dagger$, such as TRBA-cr, we reimplement it with the same training data, but the training configuration follows the official. CRNN-ViSu and TRBA-ViSu use the same training data as ViSu. As detailed in Appendix B.1, they both adopt OGS, CUA loss, URF, and a 0.5 probability of rotation by 180 degrees. The batchsize is set to 384 for both synthetic data and real unlabeled data.

---

> > ### Comment · Reviewer_qi8n · 2024-08-12
> >
> > Thank you for your effort on the response. All my concerns have been properly addressed.  I also reviewed the comments from the other reviewers, and I think most of the concerns have been addressed. Additionally, I suggest incorporating the evaluation improvement experiments from the rebuttal into the paper to better evaluate the model.

---

> > > ### Author Response · Authors · 2024-08-12
> > >
> > > Thanks for your response and suggestions. We will integrate these details in the final version.

---

### Official Review · Reviewer_S6t1 · 2024-07-07

**Soundness:** 3
**Presentation:** 3
**Contribution:** 3
**Rating:** 7
**Confidence:** 4

**Summary:**

This paper focuses on the character morphologies and proposes to boost the scene text recognizer through viewing and summarizing paradigm. In the viewing process, the Mean Teacher framework is used to train with unlabeled data. In the summarizing process, the proposed method theoretically proves the mistakes in the previous method and proposes a new loss function.

I have read the author responses and the comments of other reviewers, I would recommend the 'accept' score.

**Strengths:**

1. The proposed method achieves good results, especially on challenging benchmarks.
2. The paper is well written. It clearly points out the problem in the previous method and provides a detailed explanation. The proposed OGS and Mean Teacher framework effectively increase the diversity of character glyphs. The CUA loss corrects the previous error and better achieves clustering of identical characters.
3. There are lots of experiments to prove the effectiveness of the proposed method.

**Weaknesses:**

1. There is a lack of clear description in the paper on how CRNN-ViSu and TRBA-ViSu in Table 1 are set up and trained.
2. Although the proposed method achieves high average accuracy on common benchmark and Union14B-Benchmark, it is not SOTA on certain benchmarks, such as SVT, SVTP, CUTE, Curve, Contextless, MultiWords.
3. When the proposed method recognizes text with vertical reading order, how to decide whether to rotate it 90 degrees clockwise or counterclockwise.

**Questions:**

See Weaknesses.

**Limitations:**

The authors adequately addressed the limitations and potential negative societal impact

---

> ### Author Rebuttal · Authors · 2024-08-03
>
> Thank you for the thoughtful and constructive review. We hope these responses will address your concerns appropriately.
>
> > Q1: There is a lack of clear description in the paper on how CRNN-ViSu and TRBA-ViSu in Table 1 are set up and trained.
>
> A1: For a fair comparison, CRNN-ViSu and TRBA-ViSu use the same training data as ViSu, namely MJSynth, SynthText, and Union14M-U. As detailed in Appendix B.1, they both incorporate OGS, CUA loss, URF, and a 0.5 probability of rotation by 180 degrees. The batchsize is set to 384 for both synthetic data and real unlabeled data.
>
> > Q2: Although the proposed method achieves high average accuracy on common benchmark and Union14B-Benchmark, it is not SOTA on certain benchmarks, such as SVT, SVTP, CUTE, Curve, Contextless, MultiWords.
>
> A2: As shown in Table. 1, the SOTA method for SVT, SVTP and Curve is TRBA-cr*[54], which uses different traning data. When we reimplement TRBA-cr$\dagger$ with the same training data, our ViSu surpasses it by 1.5%, 1.2%, and 1.1% on SVT, SVTP, and Curve, respectively.  A similar situation occurs with the SOTA method TRBA-pr[1] for Contextless and Multi-Words, which uses real labeled and unlabeled data for training. For the CUTE benchmark, with its small sample size of only 288, our method is 1.1% (3 samples) lower than MATRN [25]. Small sample sizes can easily lead to error fluctuations and instability. However, MATRN is 0.7% to 2.7% lower than our method on other common benchmarks and Union14M-Benchmark, demonstrating the stability of our method.
>
> > Q3: When the proposed method recognizes text with vertical reading order, how to decide whether to rotate it 90 degrees clockwise or counterclockwise.
>
> A3: For text images with a vertical reading order, the model will recognize the results after rotating the image by both 90 and 270 degrees and will select the one with the higher confidence as the final recognition result.

---

### Official Review · Reviewer_Smtd · 2024-07-12

**Soundness:** 2
**Presentation:** 3
**Contribution:** 3
**Rating:** 6
**Confidence:** 4

**Summary:**

Existing scene text recognition (STR) methods struggle to recognize challenging texts, which originates from the insufficient exploration of character morphologies. To address the issues, the paper proposes to facilitate the contrastive learning-based STR framework in a self-motivated manner by leveraging synthetic and real unlabeled data without any human cost. An Online Generation Strategy is proposed to enrich the diversity of characters in training data. Besides, a new Character Unidirectional Alignment Loss is proposed for aligning the character features in the student model with the reference features in the teacher model. Extensive experiment results show the effectiveness of the proposed method.

**Strengths:**

1) The paper proposes to improve the recognition for challenging texts, especially for artistic and severely distorted characters within a mean-teacher framework without the requirement for real labeled data.
2) An Online Generation Strategy is proposed to enrich the diversity of characters in training data without need of psuedo labeling real data explicitly.
3) A Character Unidirectional Alignment Loss is proposed to improve the existing Character Contrastive (CC) Los [45] in character representation learning.
4) Extensive experiments on Common Benchmarks, Union14M-Benchmark, and other challenging benchmarks demonstrate the effectiveness of the proposed method.

**Weaknesses:**

1) The overall framework follows a mean teacher framework, which is not new for semi-supervised scene text recognition.
2) In Table 1, the baseline model without RU data has achieved a high performance than existing SOTA. Is there any special design in the baseline model?
3) Why not simply remove the second item in equation, i.e., exclude the positives in the denominator?
4) Absense of comparison between the proposed method and other methods in Sec. 2.3.

**Questions:**

Please refer to the weakness part.

**Limitations:**

Yes. The limiation is well descirbed in the paper.

---

> ### Author Rebuttal · Authors · 2024-08-03
>
> We appreciate the reviewer's constructive feedback and address each concern below.
>
> > Q1: The overall framework follows a mean teacher framework, which is not new for semi-supervised scene text recognition.
>
> A1: While certain methods, such as Zheng et al.[54], also employ the mean teacher framework for semi-supervised OCR, they do not take the character morphologies into consideration, making them suboptimal. We adopt the mean teacher framework to incorporate real unlabeled data, with our primary contribution being the proposal of a novel training paradigm through viewing and summarizing. Overall, we make two significant improvements to the mean teacher framework:
>
> (1). The conventional mean teacher framework usually only utilizes information from unlabeled data without additional exploration of labeled data. However, our method proposes the Online Generation Strategy (OGS) to further exploit labeled data. With the equipment of OGS, the viewing process enriches the diversity of character morphologies, facilitating the model to generalize the ability to recognize challenging samples using only simple synthetic training data.
>
> (2). The ordinary mean teacher framework aligns the features of two perspectives of one unlabeled sample without optimization for text-specific characteristics. Our method enhances the robustness of the model to character morphology during the summarizing process. On the one hand, Unified Representation Forms (URF) is used to unify text image reading order and character orientation. On the other hand, Character Unidirectional Alignment (CUA) loss is adopted to theoretically correct the previous formula error and obtain unified features for each character category.
>
> Benefiting from the above exploration of the monotonousness of training data and sensitivity of the model to character morphologies, our method achieves SOTA performance on all benchmarks, with particularly notable superiority on the challenging Union14M-Benchmark.
>
> > Q2: In Table 1, the baseline model without RU data has achieved a high performance than existing SOTA. Is there any special design in the baseline model?
>
> A2: Our baseline model incorporates URF. As described in lines 269-270, the same applies to CRNN and TRBA baselines. As shown in Table 1, our baseline achieves higher accuracy only on the Multi-Oriented (M-O) of Union14M-Benchmark. This is because URF is particularly effective at handling text with vertical and reverse reading orders, which are prevalent in M-O.  In the following, we further present the performance of the baseline model without URF. The experiments indicate that our ViSu equipped with URF significantly excels in recognizing challenging texts. We will also release the code.
>
> | Method | Datasets | IIIT | SVT | IC13 | IC15 | SVTP | CUTE | WAVG | Cur | M-O | Art | Con | Sal | M-W | Gen | AVG
> |  ----  | ----  | ----  | ----  | ----  | ----  | ----  | ----  | ----  | ----  | ----  | ----  | ----  | ----  | ----  | ----  | ----
> | Baseline without URF | SL | 96.1 | 94.0 | 95.9 | 85.8 | 87.8 | 89.9 | 92.3 | 58.0 | 17.0 | 52.9 | 55.4 | 64.9 | 54.8 | 61.3 | 52.0
> | Baseline | SL | 96.1 | 94.4 | 96.5 | 86.0 | 88.2 | 88.5 | 92.5 | 60.5 | 82.7 | 53.6 | 54.0 | 70.6 | 54.5 | 62.0 | 62.6
> | ViSu | SL+RU | **97.6** | **96.1** | **98.3** | **89.3** | **91.3** | **92.4** | **94.7** | **71.6** | **85.8** | **66.8** | **57.6** | **80.3** | **62.6** | **71.2** | **70.9**
>
> > Q3: Why not simply remove the second item in equation, i.e., exclude the positives in the denominator?
>
> A3: The second term in the denominator represents the feature similarity between the character and all identical characters in the mini batch except for itself. When optimizing CUA loss, the feature distances of all characters within the same category are minimized, i.e., clustering the same characters. As illustrated in Fig. 1(b), the same character can have very different representation forms. This optimization process plays a crucial role in extracting the visual feature commonalities, which can help the network enhance its robustness to character morphologies. Experimental results supporting this are presented below.
>
> | Method | Cur | M-O | Art | Con | Sal | M-W | Gen | AVG
> |  ----  | ----  | ----  | ----  | ----  | ----  | ----  | ----  | ----
> | without CUA Loss | 69.9 | 83.9 | 64.1 | 54.9 | 76.0 | 60.8 | 67.6 | 68.2
> | CUA Loss without second term | 71.1 | 85.0 | 65.3 | **57.9** | 79.5 | 62.4 | 70.6 | 70.3
> | CUA Loss | **71.6** | **85.8** | **66.8** | 57.6 | **80.3** | **62.6** | **71.2** | **70.9**
>
> > Q4: Absense of comparison between the proposed method and other methods in Sec. 2.3.
>
> A4: In Sec. 2.3, we reference Baek et al.[1], ABINet[10], Zheng et al.[54], Gao et al.[11], and Seq-UPS[27]. We compare ViSu with [1][10][54] in Sec. 4. However, [11] and [27] did not release their code or checkpoints, preventing detailed comparisons on metrics such as performance on the Union14M Benchmark, parameters, and speed. In the following, we compare the metrics reported in their papers with our method. The experimental results indicate that our ViSu significantly outperforms these methods on several common benchmarks.
>
> | Method | IIIT | SVT | IC13 | IC15 | SVTP | CUTE
> |  ----  | ----  | ----  | ----  | ----  | ----  | ----
> | Gao et al.[11] | 74.8 | 78.1 | 81.2 | 54.7 | - | -
> | Gao et al.[11] (ensemble) | 76.8 | 80.8 | 84.5 | 57.6 | - | -
> | Seq-UPS[27] | 92.7 | 88.6 | 92.2 | 76.9 | 78.8 | 84.4
> | Seq-UPS[27] w/ SeqCLR (All-to-instance) | 92.3 | 87.2 | 91.8 | 77.9 | 78.9 | 85.4
> | Seq-UPS[27] w/ SeqCLR (Frame-to-instance) | 92.8 | 86.7 | 92.6 | 77.4 | 79.2 | 86.1
> | Seq-UPS[27] w/ SeqCLR (Window-to-instance) | 93.1 | 86.7 | 91.7 | 76.8 | 81.4 | 85.9
> | ViSu | **97.6** | **96.1** | **98.3** | **89.3** | **91.3** | **92.4**

---

> > ### Comment · Reviewer_Smtd · 2024-08-12
> > **Official Comment by Reviewer Smtd**
> >
> > Thanks for the response by the authors.
> >
> > All my concerns have been properly addressed.
> > After considering the rebuttals for all the reviews, I'd like to raise the score to WA.

---

> > > ### Author Response · Authors · 2024-08-12
> > >
> > > We thank the reviewer for engaging in the discussion and updating the score. We will further revise our paper in the final version.

---

### Official Review · Reviewer_9AqC · 2024-07-18

**Soundness:** 3
**Presentation:** 2
**Contribution:** 3
**Rating:** 6
**Confidence:** 3

**Summary:**

This paper addresses the problem of insufficient exploration of character morphology in scene text recognition and proposes a new framework comprising an Online Generation Strategy (OGS) and Character Unidirectional Alignment (CUA) Loss to enable the model to learn from unlabeled real data. OGS mitigates the issue of data scarcity, while CUA aids in clustering characters. Comprehensive experiments demonstrate the effectiveness of the proposed method.

**Strengths:**

1. The proposed method effectively addresses the issue of excessive representation methods for scene images through Unified Representation Forms.
2. The errors in the previous Character Contrastive Loss method are corrected.
3. The idea of utilizing character morphology is novel.

**Weaknesses:**

1. The writing quality needs improvement.
2. There is a lack of detailed explanation regarding the Online Generation Strategy (OGS) and Unified Representation Forms (URF).

**Questions:**

The proposed method struggles with handling images with extreme aspect ratios. The authors attempt to address this by splitting the image, recognizing each part, and then concatenating them. However, this splitting operation may interfere with character recognition and degrade the overall recognition quality.

**Limitations:**

N.A

---

> ### Author Rebuttal · Authors · 2024-08-03
>
> We thank the reviewer for the constructive feedback provided. Below, we address the identified weaknesses and questions.
>
> > Q1: The writing quality needs improvement.
>
> A1: We appreciate your observation regarding the writing quality. We will refine and enhance the clarity of the writing.
>
> > Q2: There is a lack of detailed explanation regarding the Online Generation Strategy (OGS) and Unified Representation Forms (URF).
>
> A2: Thanks for highlighting this issue. We will revise these sections in the subsequent version.
>
> Unified Representation Forms: Different from typical visual objects, text images are characterized by serialized information, necessitating a specific reading order dictated by linguistic rules. Moreover, scene text images frequently exhibit rotation, resulting in varied character orientations. As illustrated in Fig. 2, the inconsistent representation forms caused by multiple combinations of reading orders and character orientations undoubtedly increase the difficulty of network convergence. Besides, because scene text images with vertical reading orders often have small aspect ratios, leading to significant deformation of visual information when resized, rendering the characters unrecognizable. To accommodate vertical and horizontal reading orders, we intuitively rotate them clockwise to unify the reading order to horizontal, which is referred to as Unified Representation Forms (URF). Specifically, an image with a height-to-width ratio exceeding a certain threshold will be rotated, otherwise it remains unchanged. It is easy to notice that the existing four representations can be further distilled into two primary forms with left-right reading order and their counterparts rotated by 180 degrees. This simplification and unification alleviate the issue of inconsistent representation forms and lay the foundation for the following Online Generation Strategy (OGS) process.
>
> Online Generation Strategy: The problem of diverse character morphology primarily stems from the simplistic and uniform training samples in synthetic datasets. Despite the abundance of samples in synthetic datasets, the homogeneity in character styles restricts the model's ability to acquire useful information for recognizing challenging texts in real scenarios. To address this problem, we propose to generate text samples in two primary forms and their 180-degree rotations based on their labels. As depicted in Fig. 2, these samples are background-free but with diverse character styles. Following the characteristics of real samples, all character orientations within a generated sample are consistent. Random selection of font, position, and character spacing enhances the diversity of character morphologies. Both synthetic training images and their corresponding online-generated samples are concurrently fed into our semi-supervised framework. By eliminating background noise and enriching the diversity of character styles, these samples encourage the model to concentrate on character morphology and generalize the ability to recognize complex texts from simple synthetic training data.
>
> > Q3: The proposed method struggles with handling images with extreme aspect ratios. The authors attempt to address this by splitting the image, recognizing each part, and then concatenating them. However, this splitting operation may interfere with character recognition and degrade the overall recognition quality.
>
> A3: (1). In practical application scenarios, text recognizers are typically attached to text detectors, working collaboratively to complete the OCR process. Therefore, employing a word-level text detector instead of a line-level detector can mitigate this issue.
>
> (2). When encountering long texts with extreme aspect ratios, resizing them to a preset aspect ratio leads to a loss of contextual information, which is a common problem among text recognizers. Although slicing long texts for recognition may indeed cause some degradation, but it still yields better results than attempting to recognize the entire text in one pass, which is a limitation of the current model. Future specialized improvements and optimizations can be developed to enhance the recognition of long texts.

---

### Decision · Program_Chairs · 2024-09-25

**Decision:**

Accept (poster)

**Comment:**

After discussion, all the reviewers provided positive feedback on this submission, therefore we recommend accepting it.